# G9a Knockdown Suppresses Cancer Aggressiveness by Facilitating Smad Protein Phosphorylation through Increasing BMP5 Expression in Luminal A Type Breast Cancer

**DOI:** 10.3390/ijms23020589

**Published:** 2022-01-06

**Authors:** Yunho Jin, Shinji Park, Soon-Yong Park, Chae-Young Lee, Da-Young Eum, Jae-Woong Shim, Si-Ho Choi, Yoo-Jin Choi, Seong-Joon Park, Kyu Heo

**Affiliations:** Research Center, Dongnam Institute of Radiological & Medical Sciences, Busan 46033, Korea; jynh33@dirams.re.kr (Y.J.); psj0516@kakao.com (S.P.); sypark@dirams.re.kr (S.-Y.P.); get3232@dirams.re.kr (C.-Y.L.); eumdayoung@dirams.re.kr (D.-Y.E.); diabolica@dirams.re.kr (J.-W.S.); sihochoi@dirams.re.kr (S.-H.C.); cyj5325@dirams.re.kr (Y.-J.C.)

**Keywords:** breast cancer, epigenetics, G9a, BMP5

## Abstract

Epigenetic abnormalities affect tumor progression, as well as gene expression and function. Among the diverse epigenetic modulators, the histone methyltransferase G9a has been focused on due to its role in accelerating tumorigenesis and metastasis. Although epigenetic dysregulation is closely related to tumor progression, reports regarding the relationship between G9a and its possible downstream factors regulating breast tumor growth are scarce. Therefore, we aimed to verify the role of G9a and its presumable downstream regulators during malignant progression of breast cancer. G9a-depleted MCF7 and T47D breast cancer cells exhibited suppressed motility, including migration and invasion, and an improved response to ionizing radiation. To identify the possible key factors underlying these effects, microarray analysis was performed, and a TGF-β superfamily member, BMP5, was selected as a prominent target gene. It was found that BMP5 expression was markedly increased by G9a knockdown. Moreover, reduction in the migration/invasion ability of MCF7 and T47D breast cancer cells was induced by BMP5. Interestingly, a G9a-depletion-mediated increase in BMP5 expression induced the phosphorylation of Smad proteins, which are the intracellular signaling mediators of BMP5. Accordingly, we concluded that the observed antitumor effects may be based on the G9a-depletion-mediated increase in BMP5 expression and the consequent facilitation of Smad protein phosphorylation.

## 1. Introduction

Breast cancer is a malignant tumor that develops around breast tissue and has a high incidence and mortality among women [1]. With approximately 2.3 million newly diagnosed cases in 2020, breast cancer has become the most common type of cancer worldwide [2]. Currently, this cancer type is considered manageable due to advancements in breast cancer treatment; however, more than 0.5 million patients lose their lives due to this malignancy annually [3]. To decrease the risk of breast cancer, a comprehensive understanding of its metastatic spread and recurrence is essential.

Breast cancer development is frequently driven by epigenetic alterations, such as DNA methylation, microRNA expression modulation and histone modifications [4,5,6,7]. These epigenetic changes may influence protein-synthesis patterns and result in cancer development by inhibiting the tumor-suppressor gene and enhancing oncogene expression [8]. Therefore, considering the importance of epigenetic regulators in tumorigenesis, discovering the key epigenetic targets for cancer therapy is essential. Among the various epigenetic modifiers, G9a has been suggested to exert oncogenic effects in multiple types of cancer [9]. This histone methyltransferase is responsible for mono- and di-methylation of histone H3 lysine 9 (H3K9), which is generally followed by transcriptional repression of target gene expression [10,11]. G9a transfers methyl groups from S-adenosyl-L-methionine (SAM) to the ε-amino group of the target lysine residue, resulting in mono- or di-methylation of K9 [12]. As epigenetic dysregulation is a well-defined characteristic of cancer cells, the histone methyltransferase G9a appears to be deeply involved in tumorigenesis [11]. Indeed, it has been documented that augmented G9a expression and the resultant increase in methylation are associated with poor prognosis [13]. A reduced G9a level is correlated with decreased expression of E-cadherin in breast cancer cell lines [14]. That is, G9a mediates the epigenetic regulation of genes related to epithelial–mesenchymal transition (EMT), leading to the suppression of breast tumor growth and metastasis. Thus, we aimed to study the breast-tumor-regulatory roles of G9a.

Bone morphogenetic proteins (BMPs) are multifunctional growth-factor proteins in the transforming growth-factor-β (TGF-β) family [15]. Initially discovered because of their roles in bone formation, they have also attracted interest due to their comprehensive functions across bodily organs [16]. They have increasingly gained attention because of their regulation of multiple cellular processes [17,18]. The biological effects of BMPs are mediated by the cytoplasmic protein family Smad, a major mediator of the TGF-β family. [19,20]. Among BMPs, BMP5 has been reported to play tumor-suppressive roles [21]. Indeed, BMP5 treatment was discovered to suppresses the proliferation and reduce the viability of adrenal carcinoma cell lines [22]. In particular, BMP5 appears to be closely associated with breast cancer. BMP5 represses the TGF-β-induced expression of Snail. In other words, as Snail is the master regulator of EMT, BMP5 seems to be able to suppress TGF-β-induced EMT. Similarly, low BMP5 levels were found in breast cancer patients and correlated with cancer recurrence [23]. Thus, we aimed to identify the presumable relationships between G9a and BMP5, suggesting that they are prominent therapeutic targets. In this study, the relationship between G9a depletion and malignant progression of breast cancer was concretely discovered. Furthermore, we suggested that BMP5 is a novel target of G9a, which is responsible for modulating the growth and metastasis of breast cancer cells.

## 2. Results

### 2.1. G9a Expression Is Associated with Poor Survival Outcomes in Breast Cancer Patients

High G9a expression appears to be closely associated with the survival rate of breast cancer patients [24]. Indeed, according to the data from Molecular Taxonomy of Breast Cancer International Consortium (METABRIC), the survival rate of breast cancer patients was reduced in the group with higher G9a expression (Figure 1A). To confirm the expression pattern of G9a according to the type of breast cancer, we assessed the level of G9a expression in several different types of breast cancer cell lines. The expression level of G9a was elevated in breast cancer cell lines compared to the normal cell line, but no type-dependent differences were found among these breast cancer cell lines (Figure 1B). As we intended to mainly focused on the changes following G9a depletion, MCF7 and T47D were used in the following experiments.

### 2.2. Loss of G9a Attenuates the Aggressiveness of Breast Cancer Cells

To identify the functions of G9a, we stably knocked down G9a gene expression in the MCF7 and T47D cell lines (Figure 2A). These G9a knockdown cell lines showed reduced migration and invasion abilities (Figure 2B). Additionally, G9a downregulation in MCF7 and T47D cells increased the expression of epithelial markers, including E-cadherin, β-catenin and Zo-1 (Figure 2C). Furthermore, an in vivo study showed decreased tumorigenesis in mice with mammary fat pad injection of G9a knockdown MCF7 cells (Figure 2D). Similarly, G9a knockdown in MCF7 cells reduced the number of not only primary spheres, but also secondary spheres, which represent a prominent characteristic of cancer stem cells (CSCs) (Figure 2E). The efficiency of ionizing radiation was also improved by G9a downregulation, as dose-dependent survival rates were increased in G9a knockdown MCF7 and T47D cells (Figure 2F). In addition, doxorubicin-mediated apoptotic cell death was dramatically enhanced in G9a knockdown cells compared with control cells, as evidenced by the significant increases in PARP and Caspase-7 cleavage (Figure 2G). Taken together, these findings suggest that loss of G9a reduces the aggressiveness of breast cancer cell lines.

### 2.3. Repression of G9a Induces BMP5 Expression

Microarray analysis was performed to identify the regulatory factors responsible for suppressing the aggressiveness of G9a-knockdown-derived tumors, and 17 genes with an expression change of at least twofold were identified (Figure 3A). Among the top 10 genes with the most pronounced differential expression, BMP5, LINC00052, and CDR1 showed increased expression, but HLA-DRA and SLC9A4 were downregulated in both MCF7 and T47D G9a knockdown cells (Figure 3B). H3K9 methylation, which is regulated by the histone methyltransferase G9a, is involved in transcriptional repression [25]. Accordingly, we focused on the genes whose expression was upregulated following G9a knockdown. We selected BMP5 as a putative target gene, since it has recently been reported to impede the migration/invasion of colorectal cancer cells and act as a tumor suppressor [21]. To identify whether G9a knockdown leads to an increase in BMP5 expression, qPCR and Western blot analysis were conducted. G9a knockdown cells exhibited augmented BMP5 expression (Figure 3C). Then, G9a occupancy at the promoter region of BMP5 was analyzed by ChIP to verify whether G9a knockdown directly causes this increase in BMP5 expression. G9a knockdown cells exhibited significantly reduced G9a occupancy at the BMP5 promoter region (Figure 3D). Moreover, reduced H3K9me2 occupancy at the promoter region of BMP5 following G9a knockdown was confirmed via ChIP assay (Figure 3D). These results indicate that G9a depletion may induce augmented expression of BMP5 and that this phenomenon is directly regulated by G9a.

### 2.4. BMP5 Reduces the Migration and Invasion Capabilities of Breast Cancer Cells

A survival analysis was performed to investigate the association between the expression level of BMP5 and patient survival outcomes. Breast cancer patients with low BMP5 expression showed poor survival outcomes compared to those of patients with high BMP5 expression (Figure 4A). This result is remarkable, since low G9a expression was implicated in low survival rates (Figure 1A). Additionally, the IHC results showed that the BMP5 level in patients with stage 3 disease was lower than that in patients with stage 2 disease (Figure 4B and Appendix A). However, no grade-dependent differences were found (data not shown). Then we deliberately modulated the BMP5 level and monitored the migration/invasion of cancer cells. Treatment with recombinant BMP5 reduced the migration/invasion capabilities of breast cancer cells (Figure 4C). Conversely, the migration/invasion capabilities were increased after downregulation of BMP5 gene expression by BMP5-siRNA (Figure 4D). Thus, these results suggest that BMP5 expression is closely related to the poor prognosis of breast cancer and plays a role in breast cancer metastasis.

### 2.5. Knockdown of G9a Induces Smad Phosphorylation through BMP5 Activation

BMPs regulate cellular processes, including cell proliferation, differentiation, migration and apoptosis [20]. These diverse roles of BMPs are mediated by their binding to transmembrane receptors and consequent phosphorylation of Smad proteins [26]. Thus, whether the G9a-knockdown-induced increase in BMP5 expression influences Smad1/5/9 phosphorylation in breast cancer cells needs to be elucidated. Increased BMP5 expression resulting from G9a knockdown did not affect the total level of either Smad1 or Smad5 (Figure 5A,B). However, G9a knockdown cells exhibited dramatically increased Smad1/5/9 phosphorylation (Figure 5B). Similarly, the increase in Smad1/5/9 phosphorylation was confirmed repeatedly by ICC (Figure 5C). To verify whether this increase in Smad1/5/9 phosphorylation is related to BMP5, changes in Smad1/5/9 phosphorylation were monitored following treatment with recombinant BMP5. Smad1/5/9 phosphorylation was increased after treatment with recombinant BMP5 (Figure 5D). Therefore, these results suggest that the increased BMP5 expression due to G9a knockdown may result in Smad1/5/9 activation in breast cancer cells.

## 3. Discussion

Cancer is characterized by epigenetic alterations, including DNA methylation, microRNA expression and histone modification. These modifications influence chromatin structure and gene expression. Indeed, the epigenetic dysregulation observed in various cancers is related to tumorigenesis [11,14,27]. In particular, the histone methyltransferase G9a has attracted attention due to its role in the initiation and progression of breast tumors. Generally, an increase in G9a expression is implicated in increasing methylation levels and promoting activation of the DNA methylation machinery in cancer [28]. In this context, we focused on the role of G9a in breast tumorigenesis.

To study the function of G9a during malignant progression, stable G9a knockdown cell lines were required. As the data in Figure 1B showed the comparatively high expression level of the G9a gene in the MCF7 and T47D cell lines, G9a gene expression was knocked down in these two cell lines, and stable G9a knockdown cell lines were established (Figure 2A). The major cause of cancer mortality is metastasis, which accounts for approximately 90% of cancer deaths [29]. Since metastasis is a process that involves cancer cell migration and invasion [30], migration and invasion assays were performed to investigate the role of G9a in breast cancer metastasis. Reduced migration and invasion abilities were observed in the G9a knockdown cell lines (Figure 2B). That is, suppression of G9a gene expression presumably prevents metastatic cancer growth.

EMT is considered a hallmark of cancer development and metastasis [31,32]. Numerous studies have shown the activation of the EMT program in breast CSCs [33,34,35]. During EMT, epithelial cells lose their properties and gain mesenchymal characteristics [36]. In detail, EMT is accompanied by loss of intercellular adherent junctions and apical–basal polarity, gain of migratory/invasive features and acquisition of a spindle-shaped cell morphology [37]. Thus, a decrease in epithelial marker expression with a concomitant increase in mesenchymal marker expression indicates EMT initiation. To identify the changes in EMT in breast cancer cells, the expression of EMT markers in G9a knockdown cells was investigated by using Western blotting. G9a downregulation increased the expression of epithelial markers in MCF7 and T47D cells (Figure 2C). Interestingly, two protein bands for E-cadherin were seen in both MCF7 and T47D cells; one for 120-kD cell surface glycoprotein, and another for previously reported 80-kD extracellular tryptic fragment of E-cadherin [38,39]. Similarly, MCF7 demonstrated one band indicating full-sized β-catenin, and the second one representing a smaller fragment of β-catenin, which is suggested to be predominately found in cancer cell nucleus [40].

Breast CSCs are widely reported to contribute to cancer progression, recurrence, drug resistance and radioresistance [41,42,43]. To identify CSCs based on their capacities, including those of self-renewal and differentiation, a sphere-formation assay was performed, since this assay has been suggested as a functional approach to study adult stem cells [44,45]. Compared to wild-type MCF7 cells, G9a knockdown MCF7 cells formed a decreased number of spheres (Figure 2E). Similar results were found when MCF7 cells were injected into the mammary fat pads of mice: tumor growth was suppressed in G9a knockdown MCF7 xenograft mice compared to wild-type MCF7 xenograft mice (Figure 2D). Furthermore, G9a knockdown not only increased the radiosensitivity of MCF7 cells but also facilitated their apoptotic death (Figure 2F,G). Taken together, these results indicate that G9a is related to the malignant progression of breast cancer cells, which might be regulated by CSCs. Interestingly, our microarray analysis indicated that BMP5 is a putative target gene regulating G9a-induced tumor aggressiveness (Figure 3A); however, few studies have documented the relationships between G9a and BMP5 in breast cancer. Therefore, we aimed to identify G9a-mediated changes in BMP5 expression and the resultant regulatory role of these changes in breast cancer growth and metastasis. Intriguingly, the lower the G9a levels, the higher the survival rates of breast cancer patients (Figure 1A). Conversely, high expression of BMP5 was linked to high survival rates (Figure 4A), implying the correlation between G9a and BMP5. We found that G9a knockdown MCF7 and T47D cells had higher BMP5 expression than their corresponding wild-type cells (Figure 3C). Increased expression of BMP5 following G9a depletion tended to improve the prognosis and suppress the metastasis of breast cancer (Figure 4). These G9a-knockdown-induced increases in BMP5 seemed to be mediated through the regulation of Smad proteins, cytoplasmic proteins that mediate the multiple biological effects of BMPs [19]. BMP5 binds to its transmembrane receptor to phosphorylate Smad proteins, and the phosphorylated Smad proteins translocate into the nucleus and subsequently regulate the expression of their target genes [26]. Dysregulated Smad signaling, which may lead to deregulation of the TGF-β pathway, is frequently found in several cancer types, including breast cancer [46]. In particular, phosphorylation of Smad1/5/9 indicates BMP activity [18]. We demonstrated phosphorylation of Smad proteins following G9a depletion in both the MCF7 and T47D cell lines (Figure 5A–C). Furthermore, the treatment of recombinant BMP5 caused an increase in Smad proteins (Figure 5D). This phenomenon is consistent with the results from a previous study that showed not only BMPs-induced Smad phosphorylation for 30 min, but also a decrease in Smad phosphorylation at 120 min [47]. That is, G9a-knockdown-induced BMP5 activation and subsequent Smad phosphorylation are thought to contribute significantly to the suppression of tumorigenesis and metastasis of MCF7 and T47D cells. In summary, this study provides novel insights into the tumor-suppressive characteristics of G9a mediated by the modulation of BMP5 expression in breast cancer cell lines.

## 4. Materials and Methods

### 4.1. Cell Lines and Cell Culture

The human breast cancer cells used in this study were purchased from the American Type Culture Collection (ATCC) and incubated at 37 °C in an atmosphere containing 20% O_2_ and 5% CO_2_. MEM and RPMI-1640 medium, each containing 10% fetal bovine serum (FBS, #SH30071.03, Hyclone, Logan, UT, USA) and antibiotic–antimycotic solution (AA, #15240062, Gibco, Carlsbad, CA, USA), were used to culture MCF7 cells and T47D cells, respectively.

### 4.2. Survival Analysis

The Cancer Target Gene Screening (CTGS; http://ctgs.biohackers.net, accessed on 1 December 2021) database was used to assess the relevance between the survival outcome of patients with breast cancer and gene-expression levels. Data from the METABRIC database, which contains information for 1980 breast cancer patients, were used for analysis.

### 4.3. RNA Interference and Generation of Stable Cell Lines

Short hairpin RNA (shRNA) transfection was performed by using Lipofectamine 3000 (#L3000001, Invitrogen, Carlsbad, CA, USA) according to the manufacturer’s protocol. From Sigma-Aldrich (SHCLND-NM_025256, Sigma-Aldrich, St. Louis, MO, USA), pLKO.1 vectors containing a nonspecific control sequence (NS) or shRNA sequence specific for G9a (shG9a) were purchased. Virus particles were obtained by transfection of 293T cells with shRNA constructs targeting G9a (#1; GGA CCT TCA TCT GCG AGT ATG, #2; AGA TTG AGC CTC CGC TGA TTT) or control constructs, along with the psPAX2 and pMD2.G plasmids, using Lipofectamine 3000. Two days after transfection, virus particles were collected, and breast cancer cells were then infected with the virus particles and selected with puromycin. Small interfering RNA (siRNA) against BMP5 was purchased from Dharmacon (L-017549-00-0005, Dharmacon, Lafayette, CO, USA), and cells were transfected by using Lipofectamine 2000 (#11668019, Invitrogen, Carlsbad, CA, USA).

### 4.4. Western Blot Analysis

Cell pellets were lysed with lysis buffer (TLP-121CETi, TransLab, Daejeon, Korea) and quantified, using a Bio-Rad protein assay kit (#5000001, Bio-Rad, Hercules, CA, USA). Equal amounts of protein were separated by SDS–PAGE and transferred to PVDF membranes (#10600100, GE Healthcare, Buckinghamshire, UK). Membranes were blocked with 5% skim milk or bovine serum albumin (BSA) at room temperature (RT) for 1 h and were then incubated with primary antibodies at 4 °C overnight. Membranes were subsequently washed with TBS-T and incubated with horseradish peroxidase-conjugated secondary antibodies at 37 °C for 1 h. Blots were developed by using ECL solution (#RPN2109, GE Healthcare, Buckinghamshire, UK) and visualized by using an Amersham ImageQuant 800 (Amersham, Buckinghamshire, UK).

### 4.5. Reagent and Antibodies

Western blot analysis was performed with antibodies specific for the following proteins: G9a (#229455, Abcam, Cambridge, MA, USA), E-cadherin (#3195S, Cell Signaling, Boston, MA, USA), β-catenin (#8480P, Cell Signaling, Boston, MA, USA), ZO-1 (#5406P, Cell signaling, Boston, MA, USA), PARP (#9542S, Cell signaling, Boston, MA, USA), Caspase-7 (#9492S, Cell signaling, Boston, MA, USA), BMP5 (#13253-1-AP, Proteintech, Chicago, IL, USA), *p*-Smad1/5/9 (#13820T, Cell Signaling, Boston, MA, USA), *p*-Smad1/5 (#9516T, Cell Signaling, Boston, MA, USA), Smad1 (#6944T, Cell Signaling, Boston, MA, USA) and β-actin. Recombinant human BMP5 (#615-BMC-020, R&D Systems, Minneapolis, MN, USA) was reconstituted at 100 ug/mL in 4 mM HCl containing 0.1% BSA. The final concentration of recombinant BMP5 used in this study was 100 ng/mL.

### 4.6. Quantitative PCR (qPCR)

Total RNA was extracted from cells by using a Ribospin kit (#304-150, GeneAll Biotechnology, Seoul, Korea) according to the manufacturer’s protocol. The quantity of isolated RNA was measured by using a NanoDrop 2000 spectrophotometer (#ND2000CLAPTOP, Thermo Scientific, Wilmington, DE, USA), and 1 µg of RNA was reverse transcribed, using an iScript cDNA synthesis kit (#1706691, Bio-Rad, Hercules, CA, USA). The qPCR was performed by using a LightCycler 96 instrument (#05815916001, Roche, Basel, Switzerland). The following primers were used: G9a forward 5′-GGA GGA AGC TGA ACT CAG GAG G-3′ and reverse 5′-GAC TGA AGT CAT CAC CCA CCA C-3′; Bmp5 forward 5′-CTC TCA TCA GGA CTC CTC CAG A-3′ and reverse 5′-GGA AGC TCA CAT AGA GTT CGT GC-3′; Smad1 forward 5′-TGC CCT CAG AAA TCA ACA GA-3′ and reverse 5′-TCA GTG AAA CCA TCC ACC AA-3′; Smad5 forward 5′-CCC AGC CTA TGG ATA CAA GC-3′ and reverse 5′-TGA AAA GCT TCT CCA ACA CG-3′; and β-actin forward 5′-AGC GAG CAT CCC CCA AAG TT-3′ and reverse 5′-GGG CAC GAA GGC TCA TCA TT-3′.

### 4.7. Migration and Invasion Assays

The migration and invasion abilities of cells were assessed as described previously [48]. For both migration and invasion assays, 8 µm–pore size transwell membranes (#353097, Corning Inc., Corning, NY, USA) were placed into the 24-well ultralow attachment plates. For invasion assay, transwell inserts were coated with 10 mg/mL of Matrigel (#354234, Corning Inc., Corning, NY, USA). The lower chambers were filled with the appropriate medium containing 10% FBS. For both assays, breast cancer cells were added to the inserts. Then, after 24 h, either migrated or invasive cells were stained with 1% crystal violet and counted in 5 microscopic fields (randomly selected) per well, using ImageJ software.

### 4.8. In Vivo Study

Breeding pairs of NSG mice were obtained from Central Lab Animal Inc (Seoul, Korea). NS or shG9a cells were injected into the mammary fat pads of 8-week-old female mice (*n* = 5 per group). Seven weeks after implantation, the mice were sacrificed, and tumor volumes were estimated by weight. The animal protocols used in this study were approved by the Institutional Animal Care and Use Committee of Dongnam Institute of Radiological and Medical Sciences (DIRAMS AEC-2020-001, Busan, Korea).

### 4.9. Sphere Formation Assay

We performed the sphere-formation assay by using a Mammosphere Culture Kit (#05620, Stemcell Technologies, Cambridge, MA, USA). Cells were seeded into ultralow-attachment 24-well plates (#3473, Corning Inc., Corning, NY, USA) in serum-free MammoCult medium supplemented with heparin and hydrocortisone. One week after seeding, spheres were visualized by using an EVOS microscope (#1253460, Life Technologies, Grand Island, NY, USA). Spheres larger than 100 µm were counted.

### 4.10. Clonogenic Assay

Cells were seeded into 6-well plates and incubated at 37 °C overnight. The next day, the cells were exposed to γ-rays from a 137Cs γ-ray source (Eckert & Ziegler, Berlin, Germany) at a dose rate of 2.6 Gy/min. Two weeks after irradiation, colonies were stained with crystal violet and counted by using ImageJ software (ImageJ version 1.8. https://imagej.nih.gov/ij/ (accessed on 1 December 2021)).

### 4.11. Microarray Analysis

To identify the connection between G9a knockdown and BMP5 gene expression, nine samples from three control and six G9a gene knockdown cell lines were selected for microarray analysis. Using Affymetrix Human 2.0 ST data, we compared the expression patterns between the samples. Differentially expressed genes (DEGs) were identified as those with a greater than twofold difference in the expression level and a *p*-value of less than 0.01. Gene enrichment and functional annotation analyses with the list of significant DEGs were performed by using DAVID, ToppFUN and KEGG. R v3.6.1 was used for all data processing and visualization of DEGs.

### 4.12. Chromatin Immunoprecipitate (ChIP)

ChIP assays were performed by using a Magna ChIP kit (#17-10085, Millipore, Billerica, MA, USA) according to the manufacturer’s protocol. The following primers were used: 0.1kb forward 5′-TCA CTG TTT CAT GTT AGA TGC GTG-3′ and reverse 5′-ACA ACC CTG CTG GGA AAG AAG A-3′; 2.0kb forward 5′-TGA GTT GGA AGG GAA GGT CG-3′ and reverse 5′-CAC CAG GCA TAT AGC CTA CCT C-3′; 3.0kb forward 5′-CTA TGC TGC CCG TGA TTA CA-3′ and reverse 5′-TGC TTC AGT TCA CTC TCT TTG AG-3′. An anti-G9a antibody (#3306S, Cell Signaling, Boston, MA, USA) and anti-H3K9me2 antibody (#CS200587, Millipore, MA, USA) were used to immunoprecipitate chromatin fragments.

### 4.13. Tissue Microarray and Immunohistochemistry (IHC)

G9a protein expression in human luminal A- and B-subtype breast cancer tissue microarrays (#BR1507 and BR1508, US Biomax Inc., Rockville, MD, USA), each containing 50 samples, was evaluated by immunohistochemistry. Paraffin-embedded tissue sections were stained with an anti-G9a antibody at 4 °C, overnight, prior to incubation with the secondary antibody at RT for 1 h. Then the sections were stained with diaminobenzidine, and images were acquired by using a microscope. The percentage of stained cells was scored as follows: 0 (<10%), 1 (10–20%), 2 (20–40%), 3 (40–60%), 4 (60–80%) and 5 (80–100%).

### 4.14. Immunocytochemistry (ICC)

Cells were fixed with formaldehyde solution and permeabilized with 0.5% Triton X-100. Then the cells were blocked with PBS solution with 0.1% Triton X-100 and 3% BSA at RT. Then the cells were stained first with an anti-G9a primary antibody and then with an Alexa Fluor 488–conjugated secondary antibody (#229455, Abcam, Cambridge, MA, USA) at RT. Stained cells were visualized by using a microscope. To quantify nuclear versus cytoplasmic localization, cells were classified according to whether they are predominantly nuclear or predominantly cytoplasm. Then either nuclear-predominant or cytoplasm-predominant cells were blindly counted. After that, the nuclear/cytoplasmic ratio of *p*-Smad1/5/9 distribution was quantified.

### 4.15. Statistical Analysis

Each experiment was performed at least three times. All statistical analyses were performed by using GraphPad Prism: * *p* < 0.05, ** *p* < 0.01, *** *p* <0.001 and **** *p* <0.0001 were considered statistically significant.

## Figures and Tables

**Figure 1 ijms-23-00589-f001:**
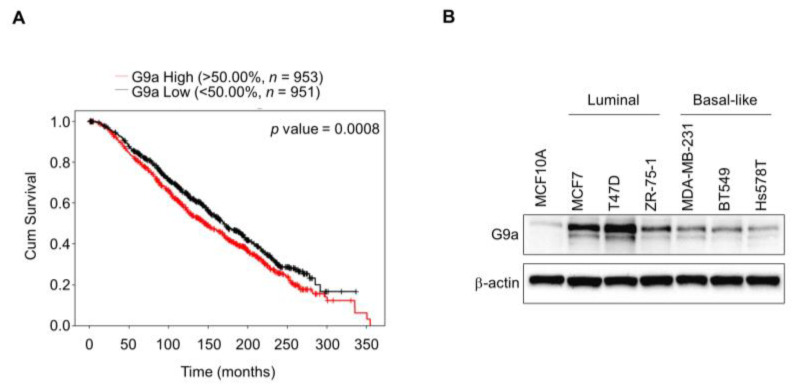
G9a expression influences survival outcomes of breast cancer patients. (**A**) Survival rate of breast cancer patients was reduced in the group with higher G9a expression. (**B**) Among several breast cancer cell lines, no type-dependent differences in G9a expression were found.

**Figure 2 ijms-23-00589-f002:**
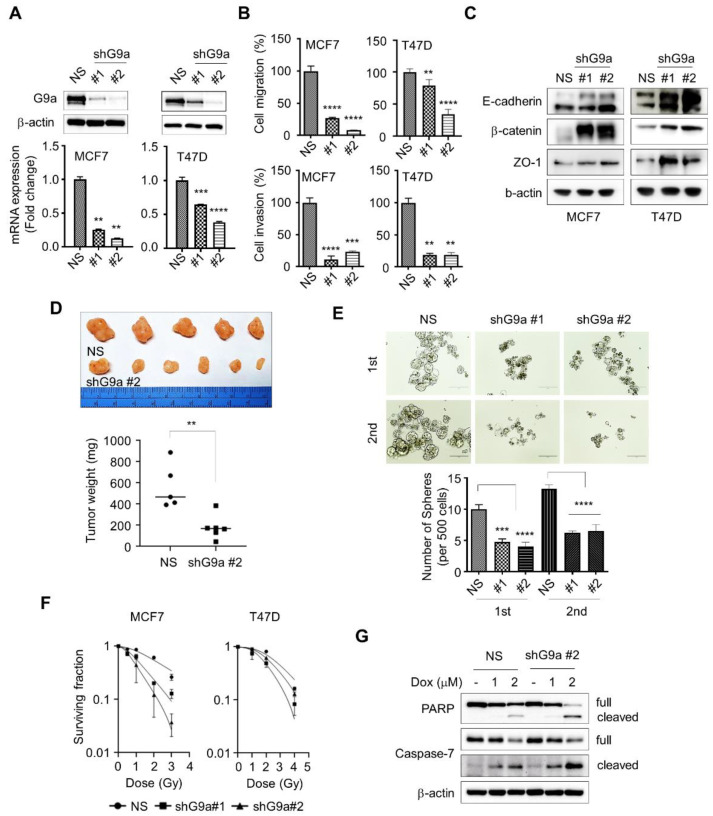
Knockdown of G9a mitigates the aggressiveness of breast cancer cells. (**A**) Expression of G9a in shRNA against G9a (shG9a) or non-silencing control (NS)-expressed breast cancer cells was evaluated by Western blot or qPCR. (**B**) G9a-depleted cell lines showed low migration and invasion abilities. (**C**) G9a-depleted cell lines showed increase in epithelial marker expressions. (**D**) In vivo study revealed reduced tumorigenesis following G9a-knockdown cell-line injection. (**E**) Sphere-forming ability was diminished by G9a depletion in MCF7 cells. (**F**) G9a downregulation enhanced the efficacy of ionizing radiation. (**G**) G9a knockdown augmented apoptotic cell death. *p* values were calculated using ANOVA. ** *p* < 0.01, *** *p* < 0.001, **** *p* < 0.0001.

**Figure 3 ijms-23-00589-f003:**
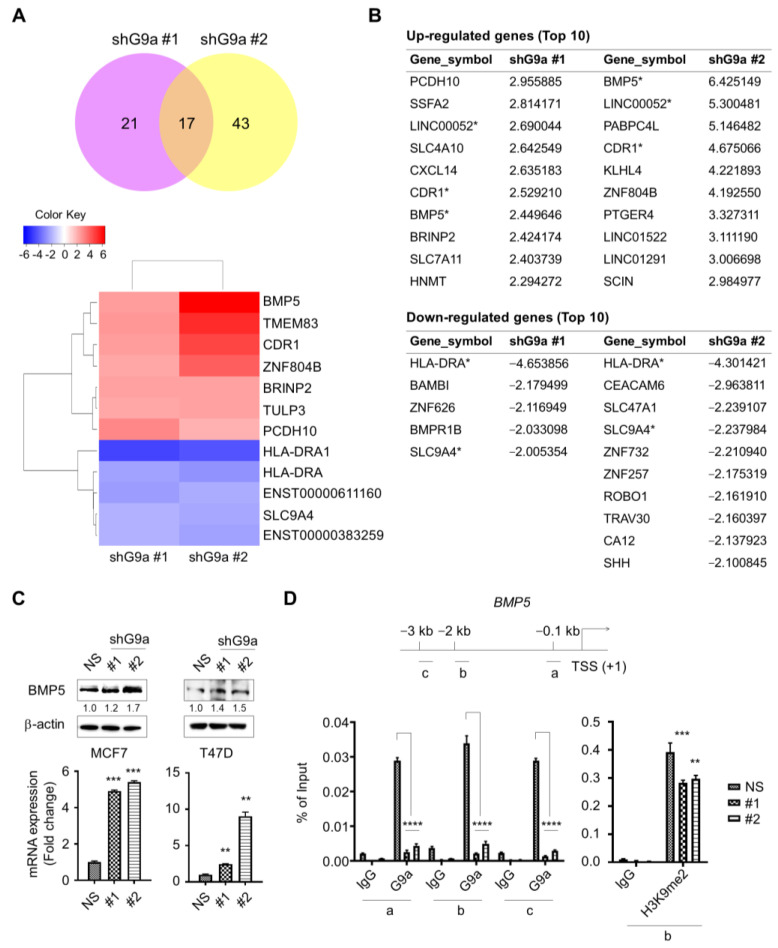
G9a knockdown may lead to increase in BMP5 expression. (**A**) Regulatory factors suppressing tumor aggressiveness were identified by using microarray analysis. (**B**) Expression levels for each gene were compared (ratios between gene expression levels of non-silencing control). In two G9a knockdown cells (shG9a #1 and shG9a #2), expression levels for each gene were compared to those of non-silencing control. Then the most upregulated and downregulated genes were screened. (**C**) G9a knockdown cells showed increase in BMP5 expression. (**D**) Reduced G9a occupancy was found at the BMP5 promoter region. G9a knockdown cells also showed decreased H3K9me2 occupancy at the promoter region of BMP5. *p* values were calculated using ANOVA. * *p* < 0.05, ** *p* < 0.01, *** *p* < 0.001, **** *p* < 0.0001.

**Figure 4 ijms-23-00589-f004:**
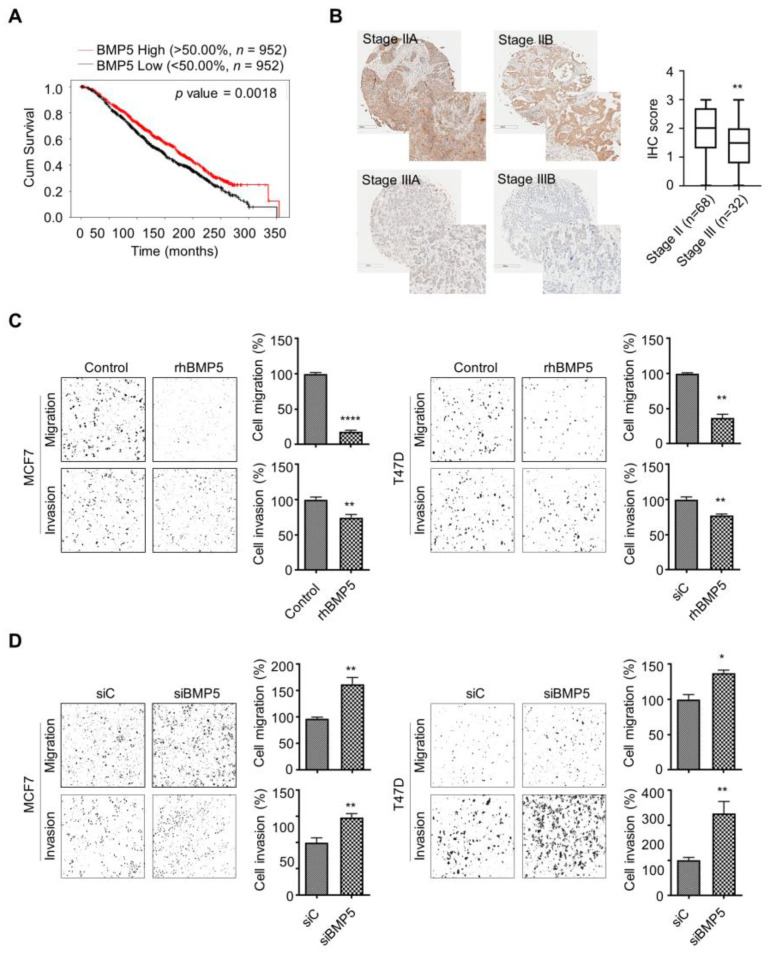
BMP5 contributes to reduce migration and invasion abilities of tumor cells. (**A**) Low BMP5 expression is associated with poor survival outcomes of breast cancer patients. (**B**) Patients with stage 3 disease showed higher BMP5 levels than those of stage 2 patients. (**C**,**D**) Migration/invasion abilities were decreased by treatment of recombinant BMP5; however, the capabilities were enhanced following BMP5 downregulation. *p* values were calculated using ANOVA. * *p* < 0.05, ** *p* < 0.01, **** *p* < 0.0001.

**Figure 5 ijms-23-00589-f005:**
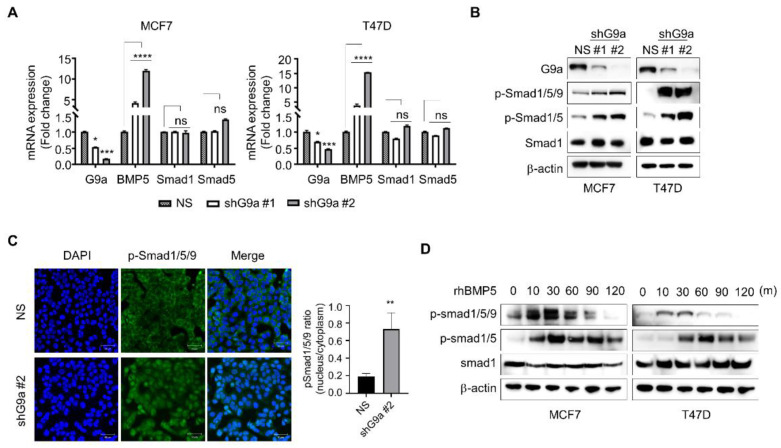
G9a knockdown facilitates Smad protein phosphorylation via BMP5 activation. (**A**,**B**) G9a-knockdown-induced increase in BMP5 expression had no effect on the total level of either Smad1 or Smad5. G9a knockdown increased phosphorylation of Smad1/5/9. (**C**) ICC demonstrated that nuclear translocation of pSmad1/5/9 was increased in G9a-depleted MCF7 cells. (**D**) Similar tendency was found after treatment of recombinant BMP5. *p* values were calculated using ANOVA. * *p* < 0.05, ** *p* < 0.01, *** *p* < 0.001, **** *p* < 0.0001, ns = not significant.

## Data Availability

Data are available upon request.

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
