# Peer review of "G9a Knockdown Suppresses Cancer Aggressiveness by Facilitating Smad Protein Phosphorylation through Increasing BMP5 Expression in Luminal A Type Breast Cancer"

_ijms, 2022, doi:10.3390/ijms23020589_

Round 1

Reviewer 1 Report

Despite the intensive research and discoveries, breast cancer is still a common cause of death for many people every year. Understanding the mechanisms leading to metastasis and recurrence is thus essential. In this manuscript the authors aim to unveil whether the epigenetic factor G9A is involved in this process, being overexpressed in breast cancer with poor prognosis. Specifically, they want to understand if a relationship exists between G9A overexpression and BMP5 low levels found in breast cancer patients.

The aim of the work is interesting, but the experimental plan is often unclear, the methods at times missing, data interpretation objectionable. Regrettably, all these make the manuscript weak.  I have some comments that the authors could address to improve their manuscript.

-The conclusive part of the abstract appears to be disconnected from the previous one and thus unclear. The authors should mention in the abstract why BMP5 overexpression explains the antitumoral effect of G9A depletion and the phosphorylation of Smad. The function of Smad should be also indicated. 

-Line 36: “Breast cancer is initiated via diverse epigenetic alterations”. The authors are probably referring to the epigenetic priming model. However, epigenetic changes would be induced by a first hit that can be genetic or environment-dependent. The authors should better clarify the sentence and/or report references supporting this statement.

-Line 76-77. The authors should mention here the database they took the data from, not only in the Methods section.

-The authors should decide how to indicate G9A, if G9A or EHMT2.

-Figure 1B: the purpose of this experiment is unclear, especially since no normal breast cell line has been used for comparison. The authors should better explain.

-The authors do not indicate how the invasive ability of the cells is evaluated and quantified. They should add this in the methods.

-Figure 2C: in the blot two bands for beta-catenin and for E-caderin are present. For E-caderin each of the two bands has a different behaviour depending on the sh used. Beta-catenin has two bands only in MCF7 cells and not in T47D. Do the authors have an explanation for that?

-Figure 3B: The authors should provide the raw data of the microarray showing the full list of genes analysed. Besides, the list in Fig. 3B shows a differential behaviour in terms of gene expression depending on the short hairpin used. The authors should comment on that in the text.

-Figure 3D: Here, the authors want to verify by ChIP “whether G9a knockdown directly causes this increase in BMP5 expression” (lines 123-124). The rationale behind this experiment is not clear. The result does show that G9A binds BMP5 promoter, but it does not give any clue whether G9A methylates or not the promoter. The authors instead conclude that “G9a knockdown cells exhibited significantly reduced G9a occupancy at the BMP5 promoter region” which is quite normal since there is no G9A to bind the promoter. The authors should revise this part. Better data would have been given by immunoprecipitating the chromatin with H3K9me(2) in G9A-depleted and -not depleted cells.

-Figure 4A: the authors could comment on the relationship between figures 1A and 4A, highlighting the eventual identity of the samples.

-Lines 140-141 “Treatment with recombinant BMP5 reduced the migration/invasion capabilities of breast cancer cells”. The authors should explain what recombinant BMP5 is and add it in the methods.

-Line 152: “BMPs regulate intracellular physiological states through Smad1/5/9 phosphorylation.” It is not clear what the authors mean. They should better explain.

-Figure 5C: From the ICC showed it would seem that p-Smad translocates from cytoplasm to the nucleus in G9A-knocked down cells, rather than globally increase. The authors maybe could couple the ICC with the quantification of the fluorescence to corroborate their conclusion “Similarly, the increase in Smad1/5/9 phosphorylation was confirmed repeatedly by ICC” (lines 157-158).

-Figure 5D: The authors claim that “Smad1/5/9 phosphorylation was increased after treatment with recombinant BMP5” (line 160). This is clear up to 30-60 minutes treatment with recombinant BMP5. However, after this timepoint the phosphorylated Smad decreases. The authors should comment on this.

Author Response

We appreciate the insightful and constructive comments from the reviewer and the considerable effort expended in the review of our manuscript. We have carefully considered each comment and suggestion of the reviewer to revise the manuscript.

Reviewer 2 Report

This is an interesting work where the role of histone methyltransferase G9a on the aggressiveness of hormone-dependent breast cancer (luminal A, ER/PR+). The authors show that G9a is correlated with poorer survival of breast cancer patients (in public database). Also, G9a knockdown leads a reduction in the ability of two cell lines (MCF7 and T47D) to migrate and invade and generate primary mammospheres. Additionally, G9a knockdown was able to induce the expression of epithelial markers, enhanced cell death in combination with radiation therapy, and promoted the expression of apoptosis-inducing factors. The effects of G9a knockdown were also reflected by a reduced tumor volume in a xenograft model of derived from MCF7cells. Microarray analysis, and further validation by Western blot, demonstrated BMP5 to be upregulated after G9a knockdown. By immunohistochemistry, BMP5 was found lower expressed in stage 3 human breast tumors compared with stage 2. The tumor suppressor role of BMP5 was then verified by ectopic treatment wither with the recombinant protein or siRNA-BMP5, and it was seen a reduction and enhancement of tumor cell invasion and migration, respectively. Finally, the authors found that after G9a knockdown, G9a occupancy of BMP5 promoter was decreased, as well as an increased phosphorylation status of Smad1/5/9 (mediators of BMP5 physiological activities).

Some major concerns must be addressed by the authors:

  1. This work is entirely carried out in MCF7 and T47D cells, which are ER and PR+ and luminal A breast cancer cells. However, the title of the manuscript talks about breast cancer, and given that there are several subtypes of breast cancer, other subtypes (i.e., triple negative) should also be assayed. Otherwise, please, be less ambiguous and specify in title (and the entire manuscript) that this work ONLY applies to ER/PR+ luminal A breast cancer.
  2. Figure 2C: mesenchymal markers (i.e., Snail, Slug) must included to decipher whether G9a knockdown is able to reverse epithelial-to-mesenchymal transition. And discuss.
  3. Figure 2E: include secondary mammospheres. And discuss.
  4. Figure 2G: show an apoptosis experiment (Annexin V by flow cytometry) to support your findings on PARP1 and Caspase 7 expression by Western blot. And discuss.
  5. Figure 5C: quantify the higher phosphorylation observed, because it is difficult to determine visually, and specify in results that it is nuclear expression (compared cytoplasmatic distribution in NS cells).

Minor revision:

  1. Indicate the meaning of “NS” in materials and methods and/or figure legends. It is supposed to be Non-Specific, but it should be clearly stated.
  2. Figure 5 legend: Fix ©

Round 2

Reviewer 1 Report

The manuscript has been significantly improved after revising. The new experiments have clearly added to the contribution of this paper, and most of my concerns from my previous review have been addressed. I have no additional comments.

Reviewer 2 Report

The authors have addressed the comments.